# Radionuclides for Targeted Therapy: Physical Properties

**DOI:** 10.3390/molecules27175429

**Published:** 2022-08-25

**Authors:** Caroline Stokke, Monika Kvassheim, Johan Blakkisrud

**Affiliations:** 1Department of Physics and Computational Radiology, Division of Radiology and Nuclear Medicine, Oslo University Hospital, P.O. Box 4959 Nydalen, 0424 Oslo, Norway; 2Department of Physics, University of Oslo, Problemveien 7, 0315 Oslo, Norway; 3Division of Clinical Medicine, University of Oslo, Problemveien 7, 0315 Oslo, Norway

**Keywords:** radionuclide, targeted therapy, radionuclide therapy, radioactivity, molecular radiotherapy, beta, alpha, auger

## Abstract

A search in PubMed revealed that 72 radionuclides have been considered for molecular or functional targeted radionuclide therapy. As radionuclide therapies increase in number and variations, it is important to understand the role of the radionuclide and the various characteristics that can render it either useful or useless. This review focuses on the physical characteristics of radionuclides that are relevant for radionuclide therapy, such as linear energy transfer, relative biological effectiveness, range, half-life, imaging properties, and radiation protection considerations. All these properties vary considerably between radionuclides and can be optimised for specific targets. Properties that are advantageous for some applications can sometimes be drawbacks for others; for instance, radionuclides that enable easy imaging can introduce more radiation protection concerns than others. Similarly, a long radiation range is beneficial in targets with heterogeneous uptake, but it also increases the radiation dose to tissues surrounding the target, and, hence, a shorter range is likely more beneficial with homogeneous uptake. While one cannot select a collection of characteristics as each radionuclide comes with an unchangeable set, all the 72 radionuclides investigated for therapy—and many more that have not yet been investigated—provide numerous sets to choose between.

## 1. Introduction

Therapies with radioactive nuclides have been rapidly increasing in both number and variations over the last few years. Treatments based on beta-minus-emitters (called beta-emitters from hereon) have been in use since the 1930s, and the first two radionuclides used for treatment purpose were sodium-24 (^24^Na) and phosphorus-32 (^32^P), both for the first time in 1936 to treat haematological disease [1]. Iodine-131 (^131^I) was the most commonly investigated beta-emitter for long, also due to the direct targeting properties of ^131^I, which allow for accumulation in differentiated thyroid cells [2]. The later emergence of carrier molecules to which radionuclides could be conjugated introduced new radionuclides in targeted therapy, such as yttrium-90 (^90^Y) and lutetium-177 (^177^Lu). While alpha-emitters have been used for decades, they have recently increased in popularity as their short range unlocks potential for tailored treatment of smaller structures [3]. Together with the currently less frequently used auger-emitters, beta- and alpha-emitters constitute the armament for therapies with radionuclides. 

The latest collection of radionuclides tabulated by the International Commission on Radiological Protection (ICRP) lists 1252 radionuclides [4]. Besides the general choice of type of radiation emitted (Table 1), there are several important factors to consider when selecting the optimal radionuclide. First, the size of the structures being targeted should be in agreement with the range of the radiation emitted. Potential heterogeneity in the uptake of the carrier molecule could also affect the optimal range; a higher degree of heterogeneity can be evened out by emitters with longer ranges. The half-life of the radionuclide should be selected with care; the pharmacokinetics of the carrier molecule will determine the optimal time for depositing the maximum amount of energy within the target tissue. The abundance of photons emitted is of importance for both radiation protection and imaging approaches. While potential radioactive daughters may add to the total amount of energy released, their possible re-localisation should be addressed. 

This review will cover the above-mentioned factors important for selection of radionuclides for targeted therapy, with emphasis on oncological applications. Historical overviews of therapeutic radionuclides both registered in the clinical trials database (https://www.clinicaltrials.gov (accessed on 30 June 2022)) and described in publications listed in PubMed (https://pubmed.ncbi.nlm.nih.gov (accessed on 30 June 2022)) are also included. Besides the physical aspects described here, factors related to radiochemistry and -pharmacy (production, cost, availability, and conjugation) are of vital importance and the reader is referred to other reviews for an overview [5,6,7]. 

## 2. Trends

A search to identify trends of publications was conducted using the PubMed database, starting with the 1252 radionuclides listed in ICRP-publication 107 [4]. This yielded a total of 72 radionuclides included in either theoretical consideration for treatment applications, preclinical, or clinical studies. To provide an overview of therapies with molecular or functional targeting mechanisms, brachytherapies with applicators were excluded in the search query, while selective internal radiation therapy (SIRT) was included (search terms are provided in Appendix A). Some publications, for example, the first ^131^I studies, have been observed missing as some of them are indexed without the specific radionuclide, but including publications without this term would have resulted in a high number of false positive hits. The timelines in Figure 1 show the publication trends over the last 74 years. A similar search to identify trends of registered clinical trials was conducted using the clinical trials database, starting with the radionuclides identified in the PubMed-search in addition to iodine-125, copper-67, and indium-111 (Figure 2). Some of the applications for the most common radionuclides are listed below.

More than 60 years after its introduction, ^131^I is still subject of an extensive number of publications. The radionuclide is commonly used to treat metastatic differentiated thyroid cancer, for ablation purposes, and for benign thyroid diseases [2,8,9]. It is also used in several conjugates; for example, the form of [^131^I]meta-iodobenzylguanidin ([^131^I]MIBG), which, due to its molecular analogy with norepinephrine, can be taken up by neuroendocrine cells. It can, therefore, be used for adult diseases, such as pheochromocytoma, paragangliomas, medullary thyroid carcinoma, and neuroendocrine carcinomas, as well as for neuroblastomas in children [10,11]. ^131^I was used in the radioimmunoconjugate [^131^I]I-tositumomab that targets the cluster of differentiation (CD) 20 antigen and was approved by the US Food and Drug Administration (FDA) for non-Hodgkin lymphoma (NHL) [12].

^177^Lu and ^90^Y have been included in more than one thousand aggregated publications each (Figure 1). Both have been explored extensively in somatostatin analogue therapy for neuroendocrine tumours, individually as well as in comparison or combination [13,14,15]. ^90^Y is used in the FDA-approved CD20-targeting radioimmunoconjugate [^90^Y]Y-ibritumomab tiuexetan to treat NHL [16]. In the last few years, SIRT for liver radioembolisation has increased rapidly, and two types of ^90^Y-based microspheres are currently approved as medical devices [17]. In addition to the FDA-approved [^177^Lu]Lu-DOTA-0-Tyr^3^-octreotate ([^177^Lu]Lu-DOTATATE) [18], ^177^Lu is part of the recently approved radiopharmaceutical used to target prostate-specific membrane antigen (PSMA) in patients with metastatic castrate-resistant prostate cancer (mCRPC), [^177^Lu]Lu-PSMA-617 [15,19]. Furthermore, ^177^Lu-based treatments are part of several clinical studies, for example, with the promising target fibroblast activation protein (FAP) [20].

The alpha-emitter radium-223 (^223^Ra) was described as one of several potential alpha-emitters for radioimmunotherapy in the late 1990s [21]. The main application has, however, been in the form of [^223^Ra]radiumdichloride, which is currently an FDA-approved treatment of skeletal metastases from castration-resistant prostate cancer [22]. Later explorations of ^223^Ra include e.g., nanomicells to increase the efficacy in treatment of osteosarcoma [23].

The beta-emitter ^32^P has had many applications since its first use against haematological diseases, including intracavitary applications and treatment of osseous metastases [24,25]. The next radionuclides in terms of aggregated research output were rhenium-186 (^186^Re), rhenium-188 (^188^Re), strontium (^89^Sr), and samarium-153 (^153^Sm), which have—together with other radionuclides—been used in palliative treatment of bone metastases or radiosynovectomy [26,27]. ^188^Re has, in recent years, been explored both for melanoma, in a radioimmunoconjugate, and for inoperable hepatocellular carcinoma, with selective administration of [^188^Re]Re-4-hexadecyl-1-2,9,9-tetramethyl-4,7-diaza-1,10-decanethiol/lipiodol [28,29]. 

The alpha-emitter astatine-211 (^211^At) was produced for the first time as early as in 1940 [30] and has been explored in numerous preclinical models. In its free state, the biodistribution is very similar to iodine’s. An up to 12-year follow-up on a cohort with ovarian cancer treated with ^211^At conjugated to MX35 F(ab’)_2_, a murine monoclonal antibody, was reported in 2019 [31]. 

Another alpha-emitter, actinium-225 (^225^Ac), has been investigated for multiple clinical uses, such as in the radioimmunoconjugate [^225^Ac]Ac-lintuzumab in treatment of acute myeloid leukemia [32]. The radionuclide is perhaps currently best known for PSMA directed treatment of mCRPC [33], and a recent meta-analysis pooled safety (225 patients) and efficacy in the form of reported response (263 patients) and outcome (200 patients) [34]. 

Some radionuclides that are either primarily or exclusively used for diagnostic purposes have been omitted from the publication trend dataset due to difficulties in separating diagnostic and therapeutic reported applications. These include copper-64 (^64^Cu), indium-111 (^111^In), technetium-99m (^99m^Tc), fluorine-18 (^18^F), gallium-68 (^68^Ga), iodine-125 (^125^I), and gallium-67 (^67^Ga). However, some of these have been investigated for explicit therapeutic applications over the years. The beta-plus- and auger-emitter ^64^Cu has, for example, been investigated for therapy in the form of a hypoxia-marker [^64^Cu]Cu-diacetyl-bis(N4-methylhiosemicarbazone) ([^64^Cu]Cu-ATSM) [35]. The gamma- and auger-emitter ^111^In was initially used to image neuroendocrine disease with somatostatin analogues but was administered with increased activity to function as treatment [36]. Another gamma- and auger-emitter, ^67^Ga, has also been reassessed as a potentially viable therapeutic radionuclide [37]. The gamma- and auger-emitter iodine-123 (^123^I) has been investigated in a pre-clinical model for treatment of glioblastoma [38]. The most common emitter in diagnostic nuclear medicine, ^99m^Tc, has, in the form of [^99m^Tc]TcO_4_^−^, actually been investigated for therapy in a preclinical breast cancer model because of its auger electron emission and interaction with the human sodium/iodine symporter [39].

## 3. Characteristics

### 3.1. Linear Energy Transfer (LET) and Relative Biological Effectiveness (RBE)

LET is a purely physical quantity, describing charged particles’ energy loss per length, measured in keV/µm. This quantity will depend on particle mass, charge, and energy, and may also vary along a particle track as the particle loses energy while traversing the material. For example, alpha-particles are known for their high LET Bragg peak at the end of the track. In general, alpha-particles have higher LET than auger electrons, which again have higher LET than beta-particles (Table 1) [40]. Auger electrons originate from electron capture or internal conversion after an isomeric transition and, hence, differ from beta-particles. They are often emitted in cascades, and will have an extremely short range, resulting in the higher LET [41,42].

Besides its close association with range, the LET is independently important for choice of emitter as it may impact the RBE. The RBE describes the effect of the radiation, for example, in terms of cell killing, and depends on both properties of the radiation and biological factors. It is defined as the biologically iso-effective absorbed doses for a certain radiation in relation to a reference radiation (often 250 keV photons). There is no established one-to-one agreement between LET and RBE, but RBE tends to increase with LET until a peak at around 100–200 keV/µm. Alpha-particles often show RBE values around five, and electrons commonly have RBE values of one (same as for reference photons). However, for auger electrons, higher RBE values have been reported when the emitters are taken up in the nucleus [41,43]. Indirect effects may kill the cells even though the emitters are not directly inserted into the nucleus, but an RBE of one, similar to other electrons, can then be expected [41,42]. While it should be emphasized that RBE values are commonly investigated in in vitro or rodent studies and are challenging to compare clinically when many additional factors contribute, it is clear that a higher efficiency per absorbed dose is to be expected from, e.g., alpha-emitters than beta-emitters. Potential benefits of this are, however, more complicated to generally interpret as both tumours and normal tissues at risk will be subject to the same effects. Differences in dose rate between the target and normal tissues, cell/tissue sensitivity, degree of oxygenation, and other radiobiological factors will impact the RBE. These should ideally be known to decide whether alpha- or beta-emitters are better suited for a specific treatment as, for instance, alpha-radiation-induced damage is independent of oxygenation, while much of the treatment effect with beta radiation requires oxygenation.

### 3.2. Particle Range

The range of a charged particle in tissue is the average distance it is expected to travel before it comes to rest (the expectation value of the path length), and, for therapeutic radiation, range depends on particle energy and particle type [44]. While many of the photons exit a patient injected with a diagnostic radionuclide for imaging, for radionuclide therapies, most of the energy should be deposited within the patient, specifically in the tissue being targeted. Hence, short-range, non-penetrating radiation; beta, alpha, and auger electrons are the ones of interest [45]. Still, the particle range ideal for a target varies greatly between cases. The range of auger electrons is very short (Table 1), and, for therapy with auger electrons to be most efficient, decays should occur in or near to the target cell nucleus [40]. Considerations of cellular uptake and intracellular positioning are, therefore, especially important for auger-emitters. Alpha-particles travel in nearly straight paths as they gradually lose energy to atomic electrons, and the range is measured from the point of decay to where ionisations stop [46]. Alphas have a much longer range than auger electrons, and it is not crucial that the decays occur near the nucleus as the alpha-particles typically traverse a few cell diameters while depositing their energy [40]. For beta-emitters, range is described differently as betas are emitted with a continuous range of energies and follow tortuous paths as they undergo multiple scatterings in a medium. Max range, mean range, and X_90_, the radius of the sphere in which 90% of the beta-emitter’s energy is deposited, can all be useful when describing range [46]. Selected properties of the 15 most common beta-emitters identified through the PubMed database search are provided in Table 2. It is important to be aware that, when range is given in literature, it usually describes the total path length travelled by the beta particle, which, in clinical settings, is longer than the depth of penetration. Beta-emitters irradiate a much larger volume than alpha-emitters, and the differences in range between beta-emitters are large enough to produce differences in therapeutic effect. 

In general, one might choose the high energy beta-emitters, such as ^90^Y, for targeting large tumours, the lower energy beta-emitters, such as ^177^Lu, for smaller tumours, and alpha-emitters for micro metastases. However, various target characteristics affect the ideal range of the therapeutic radiation; examples include size, geometry, and distribution of the molecular target. Sometimes, there can be a variety of tumour characteristics within a patient, and, in such a case, the optimal range could be different for every tumour. Several studies have investigated the impact of particle range for different tumour characteristics, such as studies by O’Donoghue et al. and Bernhardt et al. investigating the relationship between tumour size and range, and Tamborino et al. looking at the influence of radionuclide choice with heterogeneously distributed radiolabelled peptides [48,49,50]. 

#### 3.2.1. Size of Targets and Tissues at Risk 

Ideally, particle range should be optimised to irradiate the entire target volume while minimising radiation to healthy tissue. This ideal range is highly variable as target characteristics vary greatly between diseases, individual patients, and individual targets. To illustrate the effects of size and geometries with different particles and energies, we performed simulations. The method description is found in Appendix A. In Figure 3, we show the simple case of a spherical target volume with homogeneous uptake in the entire sphere. The energy deposited in the sphere as a percentage of total energy is plotted against sphere diameter for alpha sources of 5.5 MeV and for beta sources with the beta emission energy spectra of ^90^Y, ^177^Lu, and ^131^I. As the sphere gets smaller, more of the energy is deposited outside the source volume and the impact of particle type and energy is larger. ^90^Y emits beta-particles of high energies, and, already for 10 mm spheres, only approximately 62% of the emitted energy is deposited in the sphere. In contrast, the alpha-emitter deposits more than 90% of its energy within the sphere even when the sphere diameter is only 0.5 mm. Hence, if only range is of interest and there is uptake in an entire spherical tumour volume with a diameter around 0.5 mm, one might want to use an alpha-emitter rather than a high-energy beta-emitter to keep most of the energy in the tumour. 

Figure 4 illustrates sources that are spherical shells, a simplified geometry typical for large tumours with varying uptake of the radiotherapeutical due to heterogeneous vascularisation and necrotic areas in the core region [51]. In the simulations, uniform activity distribution is assumed in the shell, without any activity in the core. As the shell thickness increases, the choice of emitter becomes less significant. As an example, with a thin shell of 0.5 mm thickness and a 10 mm outer diameter, more than 95% of the energy from alpha decays is deposited in the shell and only 1.5% is deposited in the core. For the same geometry, beta radiation with the energy spectrum of ^90^Y deposits only 20% of the energy in the shell and 26% is deposited in the core. Hence, if one wishes to irradiate a volume without uptake of the radiopharmaceutical, a high-energy beta-emitter is a better choice than an alpha-emitter. How high the beta energy should be will be a trade-off between energy deposited outside the shell being kept reasonably low and sufficiently irradiating the core. If the radius of the core is assumed smaller, one can choose a lower-energy beta-emitter and still deposit sufficient energy in the core to kill the cells while limiting radiation to surrounding healthy tissue. Depending on how vital or radiation-sensitive the surrounding tissue is, in some situations irradiating the core might be less important than protecting surrounding tissues. In cases with active tumour cells without uptake in the core, ranges adequate to reach those cells from areas of uptake will likely be imperative, and, for example, ^90^Y might be the favoured radionuclide.

Optimal range is not only determined by tumour geometry and heterogeneity; uptake in healthy tissue or surrounding critical organs can be more important. For instance, with PSMA therapy for mCRPC, diagnostic scans prior to therapy reveal areas with uptake. Kratochwil et al. considered diffuse bone marrow infiltration in a patient a contraindication for treatment with beta-emitting [^177^Lu]Lu-PSMA-617 and administered [^225^Ac]Ac-PSMA-617 instead to avoid limiting hematologic toxicity [33]. Hobbs et al. presented a model illustrating the importance of accounting for micron-scale activity distributions and anatomy when calculating doses to bone marrow with alpha-emitters due to the short range of the radiation. They showed that strongly increasing the average absorbed dose to the bone marrow from ^223^Ra did not necessarily lead to toxicity since only a smaller fraction of cells were compromised [52]. Hobbs et al. similarly developed a nephron- and cellular-based model for kidneys as average dose to kidney is a poor predictor of biological response for short-range alpha radiation [53]. Hence, while short-range alpha radiation to a smaller degree exposes immediate surrounding tissue to radiation, uptake in smaller structures in healthy tissue can have large consequences due to the high amount of energy released over the short range.

#### 3.2.2. Heterogeneity 

There are many levels of heterogeneity in radionuclide therapy. There is heterogeneity between patients, between tumours, and within tumours. The most relevant in the context of particle range is heterogeneity within a tumour, specifically heterogeneous distribution of uptake of the radiotherapeutical. This can be caused, for instance, by permeability and extent of tumour vasculature or affinity of the targeting radiopharmaceutical for tumour cells [48]. For the long ranges achieved with beta radiation, much of the energy of the particle is deposited outside the accumulating cell, a mechanism termed cross-fire effect. Cross-fire effect is important for the efficacy of some radiotherapies as it can ensure radiation of tumour cells without uptake of the radiotherapeutical [40]. However, in cases where the range of the radiation is larger than the metastases, the cross-fire effect will deliver the radiation dose to surrounding tissue, causing the ratio of absorbed dose to tumour over healthy tissue to decrease [51]. Figure 4 illustrates a simplified geometry of a tumour, where a large core without uptake is found for the thinner shells. In such a case, the cross-fire effect ensures a more homogeneous absorbed dose distribution. Another typical case of uptake heterogeneity occurs when only single cells or small cell clusters accumulate the radiopharmaceutical and they are positioned at various intervals [51]. Enger et al. showed that the cross-fire effect can be an advantage in these cases as well [51]. 

### 3.3. Physical Half-Life

The physical half-life, i.e., the time it takes for the activity to reach half its initial value by radioactive decay, is inherent to the radionuclide, unchangeable by the chemical and physical environment and often known to a high degree of precision. The radionuclide with the optimal physical half-life for a certain pharmaceutical and application will depend on several factors, mainly related to the pharmacokinetics in targets and normal tissues at risk. Figure 5 illustrates some examples with normal organ and tumour tissue with different uptake and clearance.

In general, the activity in the different tissues, and, hence, the absorbed dose rate, is continually changing after administration of the radiopharmaceutical. Most radiotherapeuticals, especially those injected intravenously, will display an uptake and a washout phase. The rates and kinetics of the different phases vary between tissues. The absorbed dose deposited from activity within a tissue (the so-called self-dose) in radionuclide therapy is given by the total numbers of disintegrations occurring per mass multiplied by the energy deposited locally. This depends on both the magnitude of the uptake and the length of time the radionuclide resides in the tissue. Contributions from surrounding tissues or other organs can also add to the total absorbed dose, but these are often minor compared to the self-dose for tissues with specific uptake.

Ideally, the initial uptake phase for targets should be short so fewer disintegrations take place en route. While uptake times can be practically instantaneous for selective treatments, for antibody vectors, they can be longer, i.e., hours or days. Schemes to pre-target tumours in order to reduce the uptake time, for example, with biotin–avidin systems, have been proposed [54]. If the uptake phase for targets is sufficiently rapid, the physical half-life of the radionuclide can, in theory, be very short. In practice, production and logistics will then limit the half-life. Radiation protection concerns may also limit a very short half-life as—keeping all other factors constant—a higher amount of activity is needed to achieve the same absorbed dose if the physical half-life is decreased, and this may lead to a higher exposure rate of the surroundings. If the uptake phase for targets is longer than for normal tissues, longer physical half-lives are preferred. 

For the wash-out phase as well, the balance between the tissues is vital. If the biological wash-out of the target tissue is slower than for the relevant normal tissues, it suggests a longer physical half-life and vice versa. An additional element to consider is that the physical half-life should not be too long compared to the biological residence time in tumours. This would lead to redistribution of activity outside targets. Radiation protection considerations could also put an upper bond on the radionuclide half-life to avoid the patient from becoming a long-lived (although low-activity) radiation source. 

For example, we can consider [^177^Lu]Lu-DOTATATE-treatment of neuroendocrine tumours. Typically, wash-out from tumours and kidneys has been observed to behave exponentially with biological half-lives of the order of 287 h and 77 h, respectively [55,56]. ^177^Lu has a physical half-life of 160.8 h, resulting in effective half-lives of 103 and 52 h. If identical uptake per tissue mass is assumed, this results in a tumour to normal tissue absorbed dose ratio of 2.0. If the physical half-life was shorter, say 50 h, this ratio would be lowered to 1.4. Increasing the physical half-life, at least to a certain point, would result in an increased tumour to normal tissue ratio.

### 3.4. Imaging Properties

Post-therapy imaging of the distribution of the radiotherapeutical is essential to validate that the uptake pattern is as intended, and often also to calculate the absorbed doses to target tissue and normal organs at risk [57]. The potential for imaging in general depends on the pharmacokinetics and distribution of the radiotherapeutical, the amount of photons produced by the disintegrations, and the administered activity of the radionuclide. Photon origin, yield, and energy, in addition to camera settings, will also impact the image quality and quantitative properties. Most beta- and auger-emitters will emit some gamma photons, allowing for gamma camera imaging (Table 2). For example, the in vivo distribution of ^177^Lu and ^131^I can be followed easily due to gammas with appropriate energies and yields [58,59]. Other beta-emitters are more challenging, and, e.g., ^90^Y was first believed to be difficult to image accurately. However, bremsstrahlung imaging and later also positron emission tomography (PET) were proven feasible [60], and the approaches have become widely used after SIRT [61]. Still, this treatment is localized, and systemic administered treatments using ^90^Y will be more challenging with regard to imaging as for example studies of [^90^Y]Y-ibritumomab tiuxetan and [^90^Y]Y-DOTA-DPhe^1^-Tyr^3^-octreotide ([^90^Y]Y-DOTATOC) have demonstrated [62,63]. Companion diagnostics—similar carrier molecules with diagnostic emitters attached—can be used to predict the distribution pattern of a radiotherapeutical, but potential deviations between the radiopharmaceuticals should still be investigated. Additionally, the half-lives of the diagnostic emitters need to approximate those of the therapeutic radionuclides for dosimetric purposes. 

Due to high LET, RBE, and decay chains frequently containing multiple alpha- or beta-emitting daughters (Figure 6), the amount of activity used for alpha therapy is typically much lower than for beta therapy. Although some alpha-emitters emit gamma photons suitable for imaging, the low administered activities result in poorer image statistics. In addition, the recorded photon energy spectra commonly include relatively high amounts of bremsstrahlung and X-rays; hence, quantification is often more complex and associated with larger uncertainties. However, imaging has been performed for treatments with radionuclides in the decay chains of radium-224 (^224^Ra) to lead-208 (^208^Pb) [64], thorium-227 (^227^Th) to lead-207 (^207^Pb) [65,66,67,68], and radium-225 (^225^Ra) to bismuth-209 (^209^Bi) [69,70,71,72,73]. The fourth decay chain in the same range of atomic numbers, radium-226 (^226^Ra) to lead-210 (^210^Pb) to lead-206 (^206^Pb), has somewhat unsuitable half-lives for most targeted treatments. It should be emphasized that, if imaging is only possible for a single or a few radionuclides in a decay chain, possible redistribution of daughters should be considered. For some alpha-emitters, isotopes with other characteristics can be found and companion diagnostics developed. An example includes lead-203 (^203^Pb) as a surrogate for lead-212 (^212^Pb), which is investigated for PSMA-based treatment of mCRPC [74]. With regard to imaging properties, terbium-149 (^149^Tb) can be highlighted as an alpha-emitter that also emits positrons, allowing for PET acquisition [75].

### 3.5. Radiation Protection Considerations

In general, external exposure from patients will depend on the radionuclides’ photon yield and energy, the amount of activity administered, and the pharmacokinetics of the radiopharmaceutical. The exposure limit of the public is set to an effective dose of 1 mSv/year in total by a European directive; however, it is also described that the member states shall ensure that dose constraints are established for the exposure of knowing and willing caretakers or comforters [76]. While thresholds of 1 mSv, 3 mSv, and 15 mSv are often used in this category for children, adults, and adults more than 60 years old, respectively, based on a consensus statement for ^131^I treatment [77], some variations in recommended precautions may be found between countries depending on the established limits and calculation methods. ^131^I-based treatments are traditionally associated with the highest degree of external exposure due to the high yield of photons (Table 2), and isolation measures and other constraints to limit exposure of the public, professionals, carers, and family members are common [78]. Depending on the characteristics, other beta-emitters may also impinge various radiation protection measures. Treatments such as [^177^Lu]Lu-DOTATATE can be given on both an out-patient and in-patient basis with suitable restrictions [79]. For beta-emitters with low photon yield (such as ^90^Y) and most alpha-emitters, external exposure from patients will be less of a concern. However, handling of compounds before patient administration, especially for prolonged durations (for example, for labelling) may still raise concerns [80]. For hospitals, waste management might also be a relevant factor as storing and disposing of radioactive waste can be required. 

Fluids from patients may lead to contamination of the surroundings, and excretion in urine, faeces, saliva, blood, and breast milk should be addressed. Stability of the radiopharmaceuticals, release of potential radioactive daughters, and volatile or gaseous radionuclides are important in this context. The biokinetics of a range of individual nuclides are described in a series of ICRP publications [81,82,83,84]. 

## 4. Discussion

While targeted treatment with radionuclides continues to increase in both diversity and the number of therapies performed, the field is currently dominated by three radionuclides: the beta-emitters ^177^Lu, ^131^I, and ^90^Y. This may reflect an inclination to “not change a winning team”; once radionuclides of beneficial properties have been established, the tendency may be that they are often used for novel compounds. The advantages of having well characterised radionuclides are evident with regard to imaging, production, chemistry, etc. The PubMed and clinical trials database searches performed in this work revealed that, over the years, numerous radionuclides have been either suggested or explored as options in treatments. To our knowledge, this is the first search where the starting point has been the full ICRP 107 list of 1252 radionuclides, not simply an author-defined selection.

Of the three most frequently used radionuclides, ^131^I is a candidate to be integrated directly into various biomolecules (such as with [^131^I]MIBG), and it has targeting properties of its own. This makes ^131^I an ideal radionuclide for some specific applications, such as treatment of metastatic differentiated thyroid cancer [8]. The wide variation in the iodine isotopes, both with respect to half-lives and radiation emitted, also provides excellent opportunities for companion diagnostics. For example, ^123^I and iodine-124 (^124^I) are alternatives that can be used for SPECT or PET imaging. However, the high yield of 364 keV photons often renders ^131^I less than desirable with regard to radiation protection. Both ^90^Y and ^177^Lu are radiometals that require chemical linkage to relevant carriers and are, in theory, therefore, more open to be replaced by other radionuclides. Examining the physical properties, it is clear that selection of one or the other may largely depend on the tissues to be targeted, surrounding normal structures, and normal tissue distribution and microstructure. ^90^Y emits electrons with longer ranges, which are well suited for heterogeneous structures, such as larger tumours with potentially poorly vascularised cores. ^177^Lu emits electrons with shorter ranges and will be preferred if a more homogeneous uptake is expected. An inhomogeneous distribution between different sub-structures in normal tissues, such as observed in kidneys for somatostatin analogues, may also suggest using a short-range beta-emitter [85]. 

Previous studies have suggested tailored treatments according to the individual patient’s tumour signatures and sizes, or even using a cocktail of emitters [86]. For example, peptide receptor radiotherapy (PRRT) with both ^177^Lu and ^90^Y has been explored using a combination strategy [87,88]. While ^90^Y is currently the most frequently used beta-emitter with a long range, alternatives are encouraged due to the lack of imageable signal from ^90^Y. As a last consideration on range, it should be noted that, although the physical range of radiation is well defined, it does not itself determine the space in which cells can be affected by radiation. Radiation-induced bystander effects or abscopal effects, which involve cells behaving as though they have been exposed to radiation when they have not, are not fully understood, but they contribute to the efficacy of radionuclide treatment [86,89]. 

For alpha-emitters, the situation is somewhat different. ^223^Ra has been the most frequently used radionuclide since the early 2010s. This is due to [^223^Ra]RaCl_2_ (Xofigo) treatment for castration-resistant prostate cancer, where the bone-seeking properties of radium as a calcium-analogue are exploited [22]. However, in the last few decades, several alpha-emitters have been investigated, mostly for compounds containing a linker. The choice of radionuclide is then more open. Range can be considered a less relevant factor to decide between different alpha-emitters since the energy will primarily be deposited within a radius of some cell diameters. Production, chemistry, physical half-life, and radioactive daughters will, therefore, be more important for choosing the optimal alpha-emitting radionuclide for a certain pharmaceutical. For example, radionuclides in the decay chain starting with ^226^Ra have inconvenient half-lives or chemical properties and are rarely used. Radioactive daughters are an important factor and especially relevant for alpha-emitters (Figure 6). The opportunity to produce additional radioactive nuclides in vivo can be both an advantage and a concern depending on where the surplus energy will be deposited. Especially, the recoil effect can break chemical binding with the carrier, releasing the daughter [90]. Biodistribution, pharmaceutical properties, half-lives, and other characteristics of the daughter will then be among the determining factors. For example, a historical consideration of ^223^Ra for therapy deemed the radionuclide as interesting due to the total of four alpha-particles but less than desired due to the radon-219 (^219^Rn) daughter that could redistribute as a noble gas [21]. However, after the clinical introduction, no evidence of re-distribution has been found. While alpha-particles are excellent at ensuring localised deposition of energy (Figure 3), this can also render the treatments less useful for heterogeneous tumours where cross-fire effects are desired. As described, the overall low photon flux from alpha-emitters will often pose challenges for quantitative imaging and dosimetry, which are currently required by a European directive [76].

No treatments based on auger-emitters are currently approved by the FDA or the European Medicines Agency (EMA). Many of the radionuclides familiar from diagnostic tracers (for example, ^99m^Tc, ^123^I, ^111^In) also decay by electron capture and/or undergo internal conversion after an isomeric transition, giving rise to smaller fractions of auger electrons. Over the last two decades, some registered clinical studies with auger-emitters, such as ^111^In and ^125^I, were found in our search [36,91,92], but none seem to have moved to clinical routine. While the short range of the auger electrons can spare adjacent local tissues at risk [93]—such as for the alpha-emitters—the photon irradiation can be considerable. For example, for PRRT, ^90^Y labelled somatostatin analogues gave a lower red marrow absorbed dose than ^111^In analogues as a result of the decreased photon contribution from the total body [94].

While it is relatively easy to list the various factors of importance for selection between different radionuclides, overall comparisons with clinical value are more challenging. Absorbed dose is a well-defined parameter used to quantify the total amount of energy deposited per mass; however, corrections for the time dependence, radiation type, and heterogeneity may still be needed to estimate the probabilities for toxicity or response. Especially, different types of emitters are challenging in this regard, and, for some applications, the possibility of having a selection of various radionuclides available may be beneficial. For example, PSMA targeted treatment with ^225^Ac is sometimes an option to [^177^Lu]Lu-PSMA treatment for individual mCRPC patients [95].

The concept of theragnostics has gained increasing focus over the last few years. Even with radiotherapeuticals that are possible to image themselves, companion diagnostic tracers are important for stratification purposes, for patient selection, in some cases individual treatment planning, and for response evaluation and follow-up. Selection of optimal therapeutic radionuclides should, therefore, also include considerations of the theragnostic properties and the need and possibility for diagnostic analogies.

In conclusion, selection of radionuclides for targeted therapy depends on a variety of factors. Identifying the optimal radionuclide might not be possible until after the biodistribution, pharmacokinetics, and uptake levels have been accurately described through clinical investigations. However, target sizes, expected intra-target homogeneity and uptake density, characteristics and pharmacokinetics of the radionuclide-bound carrier, administration route, expected normal tissues at risk, and other considerations may serve to guide the selection in terms of physical properties. In our search, 72 radionuclides out of a total of 1252 were found to have been considered for clinical use, and 21 were found to have been included in clinical trials. This illustrates the diversity and opportunities in the field.

**Figure 6 molecules-27-05429-f006:**
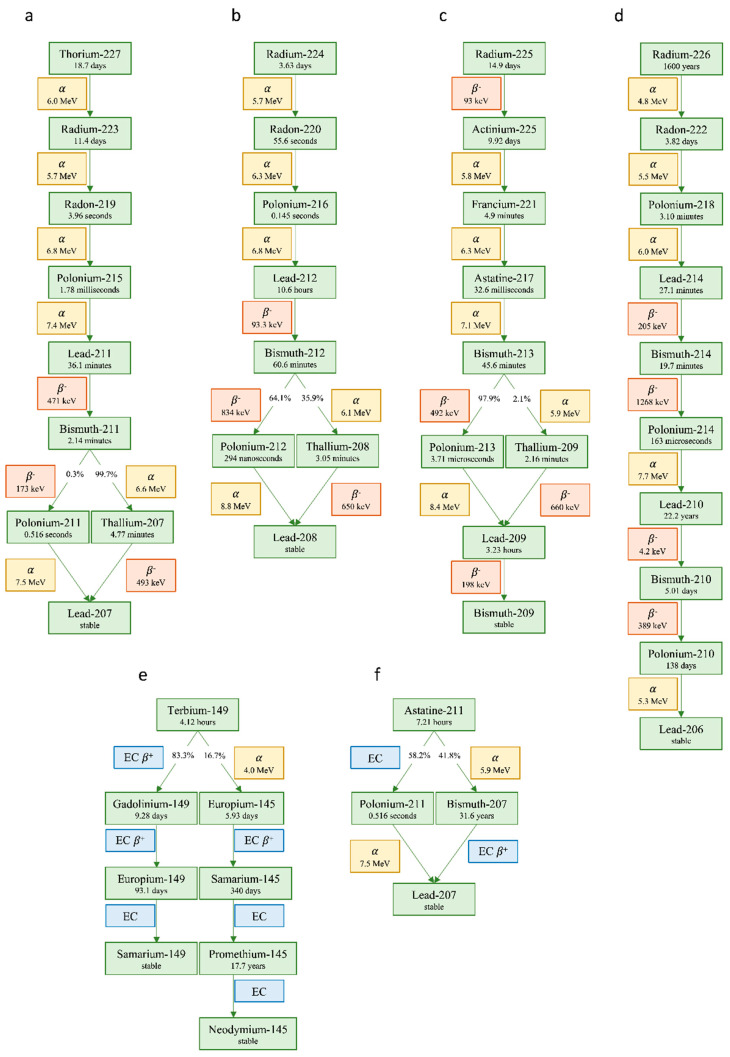
Six decay schemes (**a**–**f**), including most alpha-emitters relevant for radionuclide therapy. All branching ratios larger than 0.1% are included. The beta- and alpha-particle energies given are for the highest intensity emission. Data from Ref. [96].

## Figures and Tables

**Figure 1 molecules-27-05429-f001:**
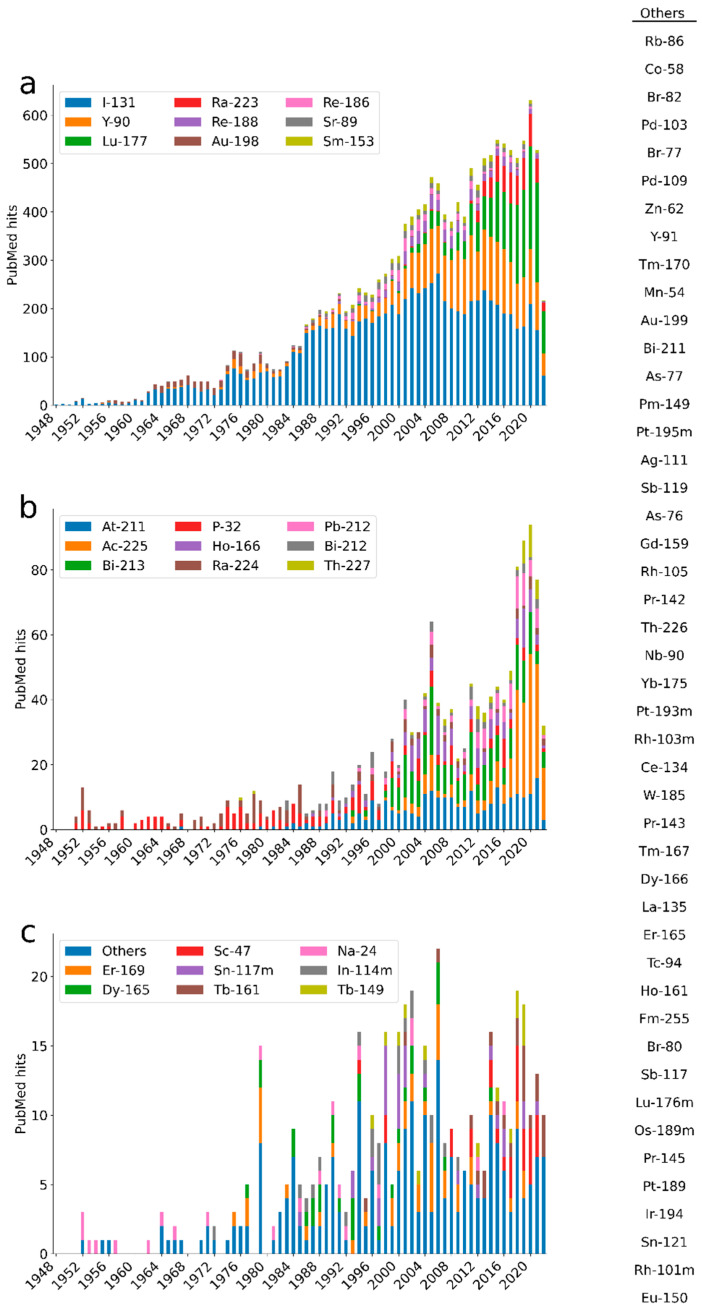
The number of hits in PubMed as of 30 June 2022, per radionuclide and year of publication. The radionuclides have been separated according to the aggregated numbers of publications across panels (**a**–**c**). The radionuclides with fewer than 13 hits total have been aggregated into “Others” and are listed to the right in the figure. The search strategy and search strings are described in Appendix A.

**Figure 2 molecules-27-05429-f002:**
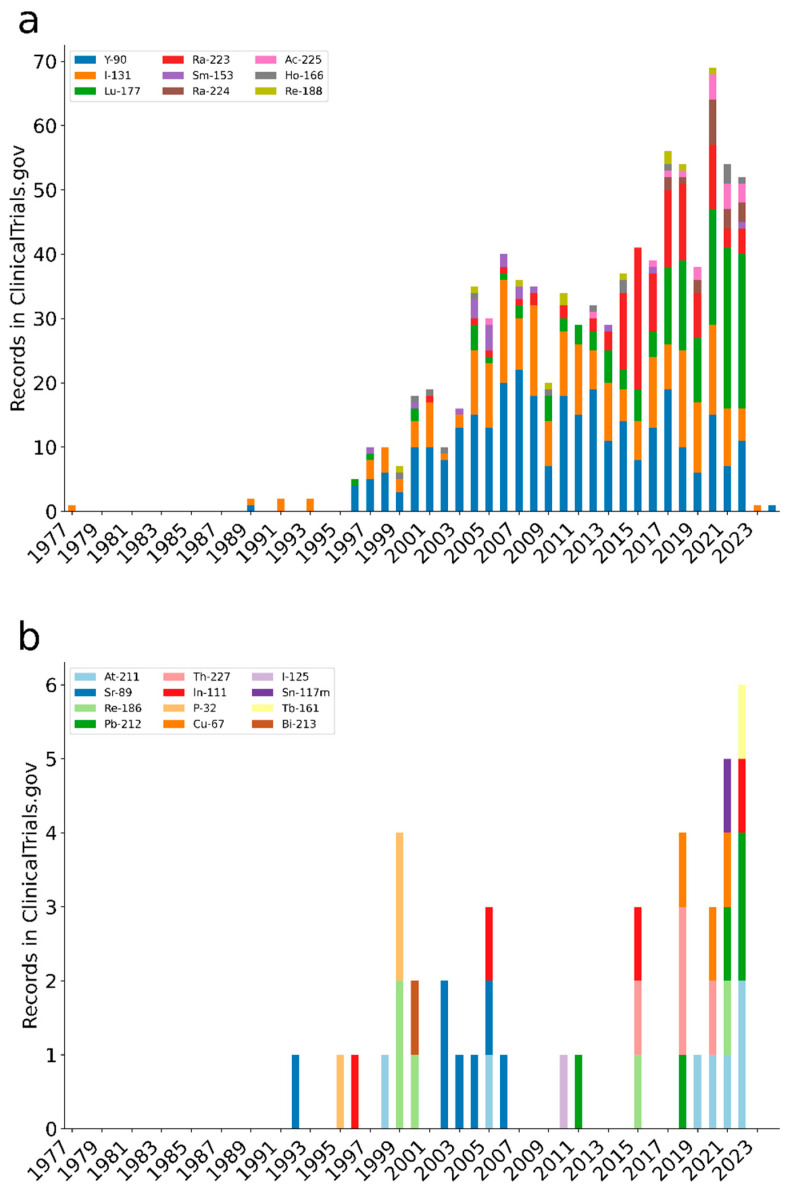
The number of records in the clinical trials database as of June 2022 per radionuclide and year. The results are split in panels (**a**,**b**) according to the aggregated number of records. The search strategy and search strings are described in Appendix A.

**Figure 3 molecules-27-05429-f003:**
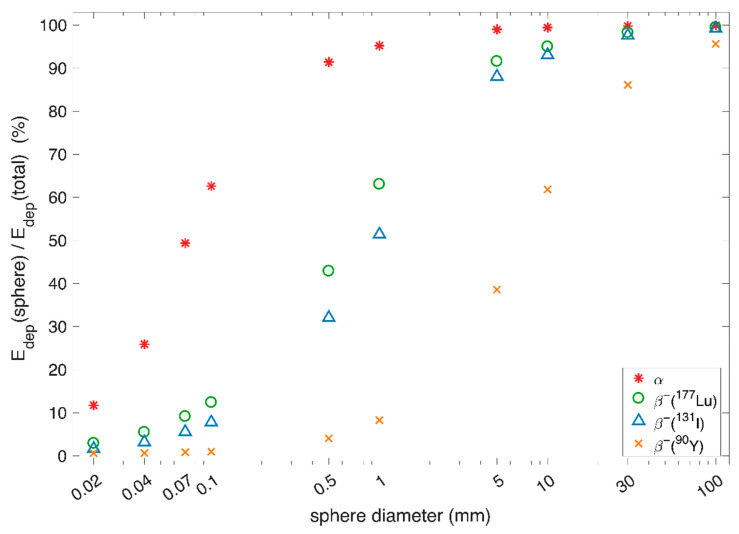
The figure shows energy deposited within a spherical source (E_dep_(sphere)) as a percentage of the total energy emitted (E_dep_(total)). The sphere source diameter ranged from 0.02 mm, to approximate a single cell, to 100 mm, to approximate a large tumour. Note that the x-axis is a log scale. Four different sources are shown, three pure beta sources with beta energies following the emission spectra for ^90^Y, ^177^Lu, and ^131^I, and an alpha source with alpha energies of 5.5 MeV. For large spheres, most of the energy will be deposited inside the sphere regardless of type of emitter.

**Figure 4 molecules-27-05429-f004:**
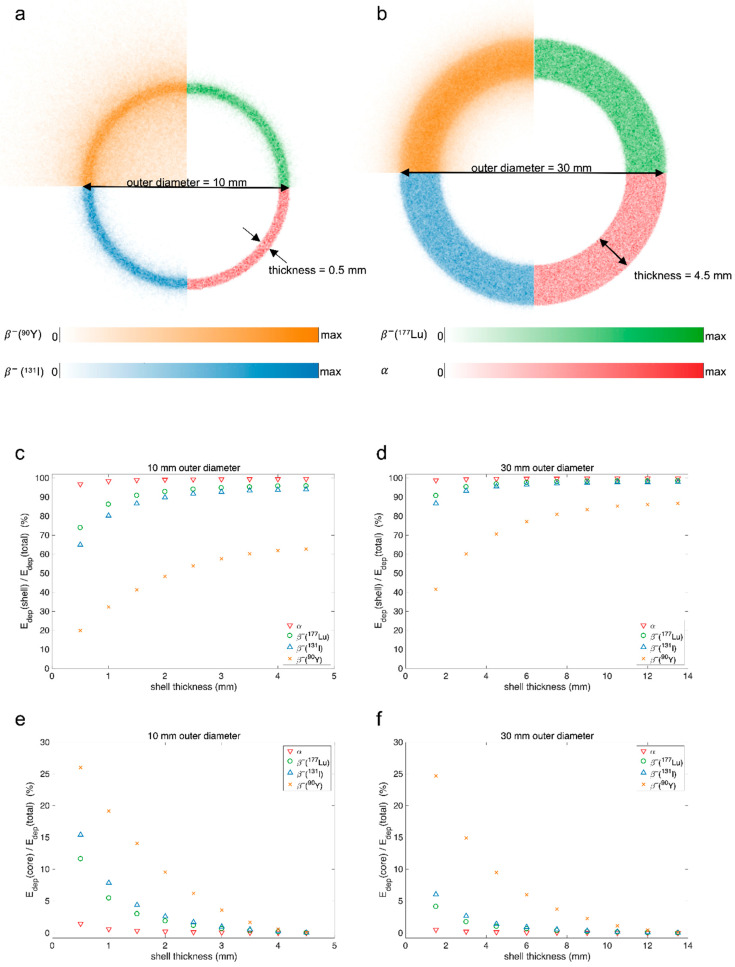
The figure shows energy deposited in and around a spherical shell source. This approximates a situation where there is uptake around the outer rim of a core without uptake. Panels (**a**,**b**) illustrate examples of shell thicknesses for the two outer diameters and show the central slices of images containing the energy deposition maps of the shells, where each quarter shows a different source. In panels (**c**,**d**), the ratio of energy deposited in the shell source (E_dep_(shell)) to total deposited energy (E_dep_(total)) is plotted against the shell thickness. In panels (**e**,**f**), the ratio of energy deposited in the core (E_dep_(core)) to total deposited energy is also plotted against shell thickness. Two different outer diameters were used: in (**a**,**c**,**e**), shells with an outer diameter of 10 mm are shown, and, in (**b**,**d**,**f**), shells with an outer diameter of 30 mm are shown. Four different sources were used, three pure beta sources with beta energies following the emission spectra for ^90^Y, ^177^Lu, and ^131^I, and an alpha source with alpha energies of 5.5 MeV.

**Figure 5 molecules-27-05429-f005:**
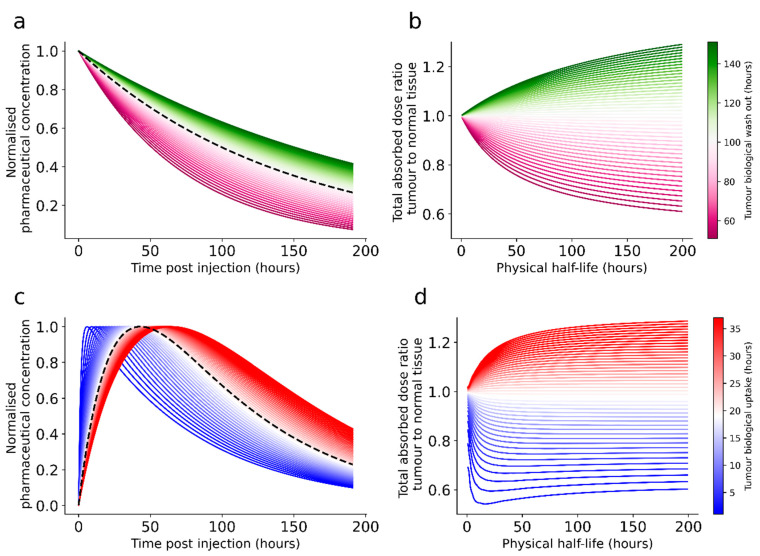
The figure illustrates theoretical situations that involve biological uptake and clearance of a radiopharmaceutical in a normal organ and tumours (**a**,**c**) and the differences in absorbed dose ratios that can be expected by selecting radionuclides with various half-lives for each (**b**,**d**). Two different types of kinetics are illustrated. In both scenarios, the normal tissue kinetics (illustrated with a dashed black line) are kept fixed and the tumour kinetics are varied. In the first situation, illustrated in panel (**a**), with the corresponding ratios in panel (**b**), an instantaneous uptake and a mono-exponential elimination is assumed for both tumour and normal tissue. The initial amount of radiopharmaceutical per tissue is set identical for both normal organ and tumours. The tumour-curves have been colour-graded according to the biological half-life, where white is equal to the normal tissue elimination (here, 100 h), whereas more saturated green indicates a slower and saturated purple indicates a faster elimination compared to the normal tissue. In panel (**b**), the ratios between the total energy absorption between tumour and normal tissue for the different tumour eliminations have been plotted for a range of physical half-lives. In panel (**c**), a different situation with bi-exponential uptake and washout is illustrated. Here, the rate of wash-out is kept fixed, while the uptake phase is varied. Again, different theoretical tumours are shown in coloured whole lines, where more saturated blue is a faster uptake and more saturated red is a slower uptake, while the normal organ is represented by a black dashed line. The curves here have been normalised to the same maximum amount of radiopharmaceutical per tissue. In panel (**d**), the absorbed dose ratios between the tumours and the normal organ are plotted over a range of physical half-lives for this scenario.

**Table 1 molecules-27-05429-t001:** Overview of types of radiation.

Type of Radiation	Particle	Mass	Typical Energy	Typical Range in Tissue	LET (keV/µm)	RBE
Alpha	2 protons and 2 neutrons	6.6 × 10^−27^ kg	Discrete; ~4–10 MeV	~20–70 µm	~50–300	~5
Beta minus	Electron	9.1 × 10^−31^ kg	Continuous; maximum of some hundred keV to some MeV	Less than a mm to some mm	~0.1–2	~1
Auger	Electron	9.1 × 10^−31^ kg	<10 keV	<1 µm	~4–26	~1 or higher

**Table 2 molecules-27-05429-t002:** According to our search, the 15 beta-emitters most frequently published on in relation to radionuclide therapy and their properties. The data are taken from ICRP 107 [4].

Radionuclide	Half-Life	Mean Energy/keV	Max Energy/MeV	X_90_ in Water ^^^/mm	Max CSDA Range in Tissue ^v^/mm	Photons Intensity > 5% and Energy > 75 keV
I-131	8.02 days	181.9	0.81	0.9	3.3	364 keV (82%) 637 keV (7%) 284 keV (6%)
Y-90	64.10 h	932.9	2.28	5.5	10.8	
Lu-177	6.65 days	133.3	0.50	0.6	1.7	208 keV (11%) 113 keV (6%)
P-32	14.26 days	694.8	1.71	3.7	8.0	
Re-188	17.00 h	762.6	2.12	4.5	10.1	155 keV (16%)
Sr-89	50.53 days	584.5	1.50	3.3	6.9	
Re-186 *	3.72 days	346.6	1.07	1.9	4.6	137 keV (9%)
Sm-153	46.50 h	223.6	0.81	1.1	3.3	103 keV (30%)
Au-198	2.70 days	312.2	1.37	1.7	6.2	412 keV (96%)
Ho-166	26.80 h	665.0	1.85	4.0	8.7	81 keV (7%)
Cu-67	61.83 h	135.9	0.56	0.6	2.0	185 keV (49%) 93 keV (16%) 91 keV (7%)
Er-169	9.40 days	99.6	0.35	0.4	1.0	
Dy-165	2.33 h	439.7	1.29	2.5	5.8	
Tb-161	6.91 days	154.3	0.59	0.7	2.1	
Sc-47	3.35 days	161.9	0.60	0.7	2.2	159 keV (68%)

* beta-particle yield of ^186^Re is 92.5%. ^^^ Calculated with Geant4 Application for Tomographic Emission (GATE); see Appendix A for details. ^v^ The continuous-slowing-down approximation (CSDA) range was calculated using the maximum energies with equation A.18 from Prestwich et al. (1989) [47].

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
