# Peer review of "Radionuclides for Targeted Therapy: Physical Properties"

_molecules, 2022, doi:10.3390/molecules27175429_

Round 1
Reviewer 1 Report
This review summarizes the progress and developments in targeted radiotherapy of cancer covering the PUBMED publications and ClinicalTrials.gov database. The authors discussed several important factors that has to be considered when selecting the optimal radionuclide from the group of alpha, beta minus and Auger emitters (Table 1). This part of the manuscript has provided interesting and comprehensive information allowing better understanding of the processes behind the action of the different types of radiation on the proliferating cells and environment.
However, there are some issues, that are listed below in no specific order that should be addressed prior to publication.
- The transcription (decoding) of all the abbreviations have to be given when they appeared for the first time in the text, including Tables.
2. The numbers of hits in PubMed is presented on Fig. 1. However among all the radionuclides 49 are assigned to category “Others”. I would suggest to provide the list of these “other” radionuclides as a table in the Supplemenatary as many of them are very promising. The references to these radionuclides (may be on the Review articles) can be also provided in the Supplementary.
3. The selected therapeutic radionuclides discussed in brief under the chapter “2. Trends”. It has to be mentioned that [177Lu]DOTATATE (Lutathera) is also approved by FDA with the corresponding reference. When compare 90Y and 177Lu the merits and pitfalls of each radionuclide need to be shortly discussed such as beta particles range in tissues, lower nephrotoxicity of 177Lu etc.
As for alpha emitting actinium-225 (Line3 130-134), the most important radiopharmaceutical based on this radionuclide is 225Ac-PSMA-617 for efficient treatment of mCRPC. The pioneer study of Heidelberg group has to be cited
5. Line 135. The authors wrote: “Some radionuclides that are primarily used for diagnostic purposes have been omitted from the publication trend data set, due to the difficulties in separating diagnostic and therapeutic applications. These include copper-64 (64Cu), indium-111 (111In), technetium-99m (99mTc), fluorine-18 (18F), gallium-68 (68Ga), iodine-125 (125I), and gallium-67 (67Ga)”. I do not think this is correct. 18F and 68Ga are positron emitting radionuclides and hardly used for radiotherapy treatment. The long-lived iodine-125 is typical isotope for brachytherapy (which is not considered as targeted radiotherapy because the radionuclide is inserted in the capsulated form, not in form of radiopharmaceutical). This sentence needs to be carefully reformulated. If the authors believe that 18F or 68Ga can be used as therapeutic isotopes, the references and examples have to be provided.
6. Also when discussing diagnostic and therapeutic radionuclides, it will be useful to mention the theranostic concept, which unavoidable part of the modern targeted radiotherapy. The authors considered some “theragnostic” radionuclides like 131I and 177Lu that have the emitting photons (gamma rays) in the decays scheme for detecting by SPECT during pre-therapeutic scan. However many therapeutic radionuclides have theranostic couples like 131I/123I (SPECT) or 131I/124I (PET), 90Y/86Y (PET) or “surrogate theranostic” couples like 68Ga/177Lu, 68Ga/225Ac. I think theranostic part is missing in the manuscript. Nowadays making 68Ga scan before 177Lu or 225Ac therapy is a gold standard in radiotherapy treatment considering higher sensitivity of PET technique.
7. Line 303. 3.3. Physical half-life. The authors wrote: “The physical half-life, i.e. the time it takes for the activity to reach half its initial value by radioactive decay, is inherent to the radionuclide, unchangeable by the chemical and physical environment and often known to a high degree of precision”. That’s really correct. In the next sentence they wrote “The optimal physical half-life will depend on several factors, mainly related to the pharmacokinetics of targets and normal tissues at risk”. What does it mean “the optimal physical half-life”? Is this type-mistake”
8. For alpha-emitters special attention has to be given to the nuclear recoil effect and its impacts on the possible use of alpha emitters for cancer treatment, proper dose estimation. At least this important effect has to be mentioned with corresponding references (for example the review articles).
9. Line 395. “An example includes lead-203 (203Pb) as a surrogate for lead-212 395 (212Pb), which is investigated for PSMA treatment [71]”. Please replace “PSMA treatment” with “mCRPC treatment”.
10. Table 2. Please add the range of beta-minus in tissue, as one of the most important characteristic.
As such, I would recommend this paper for publication after considering above questions and comments.
Author Response
-
Thank you for pointing this out. The transcription of all abbreviations are now given when they first appeared. Abbreviations used in tables have also been decoded, with the exception of radionuclides, which are decoded when used in the text only.
-
The authors agree that the radionuclides in “Others” should be listed, and a list of the radionuclides “Others” has been added to Fig. 1.
-
Thank you for the comment, we agree that this was not clearly written and have now specified that [177Lu]DOTATATE (Lutathera) is also FDA approved. While we hesitate to include comparisons in the chapter “trends”, we believe that range has been thoroughly discussed later, but has now also included an example of normal tissue differences for kidneys in the discussion.
-
Reference to the work suggested by the referee has been added to the manuscript. We have also included a brief discussion of the challenges associated with comparing different radionuclides. While we find comparing 177Lu-PSMA and 225Ac-PSMA to be very interesting, a direct discussion of which radionuclide performs best is currently difficult due to different patient groups, etc. We have however tried to describe that various emitters may be applied in different situations, and as an example included the potential benefits of 225Ac-PSMA for patients with diffuse bone infiltration in the characteristics section.
-
The sentence was not clearly written and in the previous form it was it was not correct. The sentence has been updated to reflect that the radionuclides either are exclusively or primarily used for diagnostic purposes.
-
We have now included mention of specific diagnostic nuclides in relevant paragraphs, and have also moved the paragraph where theragnostics was mentioned from “imaging properties” to the discussion -to give this important concept more emphasis.
-
The sentence has been updated to better reflect the desired meaning, namely that a certain application has an optimal physical half-life, and not that the physical half-life is dependent on the factors listed.
-
This has now been described specifically, with a reference added.
-
A replacement has been done and the sentence rewritten
-
Ranges have now been added to table 2, thank you for the suggestion. Additionally, the table was updated to only use data from ICRP 107, rather than from different references.
Reviewer 2 Report
The paper reviews the use of 73 radionuclides in clinical radionuclide therapy. The physical characteristics including linear energy transfer, relative biological effectiveness, range, half-life, and imaging properties, and radiation protection considerations are discussed to understand the complicate administration processes of radionuclides in patients. This paper is informative for the readers.
Some suggestions are noted below:
1) Line 56 better as “clinical trails (ClinicalTrail.gov) database…”. It would be better to also provide the correlated website here.
2) Figure 1 The first letter of caption in the figure should be capitalized.
3) Author should reduce the use of abbreviation if the phrase is not used often. Some abbreviations are used without being defined. Line 132: what is mCRPC? Line 444 what is MIBG?
4) Reference 38 and 43 are repetitive.
5) The representation of Figure 5 is confusing. The explanation in a-d is not clear. This section should be rewritten with indication in the figure also.
6) Line 382, what is ‘fig.y’ indicating?
Author Response
-
The URL of the website has been inserted
-
The figure is now updated with capitalized element names.
-
Thank you for the comment, we have gone through and updated the abbreviations in the text, making sure we decode them all as per reviewer 1s comment. We have only used abbreviations where we find them clarifying. We have defined mCRPC on page 6. 131I-MIBG is defined on page 6 as well and has been updated in line 444.
-
This is a regrettable error and the reference should be given only once, this has been updated in the manuscript.
-
The figure caption has been rewritten and the figure has been updated to –hopefully- be more clear.
-
Thank you for mentioning this. It is a typo, the reference has been updated.